# Oxidative Stress and Antioxidants—A Critical Review on In Vitro Antioxidant Assays

**DOI:** 10.3390/antiox11122388

**Published:** 2022-12-01

**Authors:** Raghavendhar R. Kotha, Fakir Shahidullah Tareq, Elif Yildiz, Devanand L. Luthria

**Affiliations:** 1Methods and Application of Food Composition Laboratory, Beltsville Human Nutrition Research Center, Agricultural Research Service, U.S. Department of Agriculture, Beltsville, MD 20705, USA; 2Keles Vocational School, Bursa Uludag University, 16740 Bursa, Turkey

**Keywords:** oxidative stress, antioxidants, foods, antioxidant assays limitations, challenges in correlating with health benefits

## Abstract

Antioxidants have been widely studied in the fields of biology, medicine, food, and nutrition sciences. There has been extensive work on developing assays for foods and biological systems. The scientific communities have well-accepted the effectiveness of endogenous antioxidants generated in the body. However, the health efficacy and the possible action of exogenous dietary antioxidants are still questionable. This may be attributed to several factors, including a lack of basic understanding of the interaction of exogenous antioxidants in the body, the lack of agreement of the different antioxidant assays, and the lack of specificity of the assays, which leads to an inability to relate specific dietary antioxidants to health outcomes. Hence, there is significant doubt regarding the relationship between dietary antioxidants to human health. In this review, we documented the variations in the current methodologies, their mechanisms, and the highly varying values for six common food substrates (fruits, vegetables, processed foods, grains, legumes, milk, and dairy-related products). Finally, we discuss the strengths and weaknesses of the antioxidant assays and examine the challenges in correlating the antioxidant activity of foods to human health.

## 1. Introduction

By definition, antioxidants prevent/inhibit/reduce oxidation processes [1,2,3,4,5,6,7,8,9,10,11]. Historically, the industry has used antioxidants to prevent metal corrosion and rubber vulcanization [2]. More recently, antioxidants have been used as food preservatives, lubricants, and stabilizers [2]. Allied market research shows the worldwide industrial market value for natural and synthetic antioxidants in 2015 was USD 2.9 billion and forecasted more than 50% overall growth to over USD 4.5 billion by 2022. In another study, Grand View Research forecasted that the global market for natural antioxidants is expected to be USD 4.1 billion by 2022 [12]. In each case, the functions of these antioxidants meet the strict chemical definition and are not associated with bodily functions.

In recent years, there has been considerable interest in the impact of naturally occurring phytochemicals in foods that may function as antioxidants in the human body. This interest has arisen from popular literature that defines oxidants as harmful and antioxidants as the antithesis and, therefore, healthful. There have been many reviews on all aspects of antioxidant assays and the relationship of antioxidants to health, including their activity, classification, and applications [2,3,5,10,13,14,15,16,17,18]. Apak et al. [15,16,17] published reviews on various antioxidant assays, their mechanisms, advantages, and limitations. Recently, Gulcin et al. [18] published a detailed review on in vitro antioxidant methods used for the determination of the antioxidant capacity of food constituents. Similarly, Shahidi et al. [3] and Alam et al. [13] published review articles on in vitro and in vivo antioxidant assays and their experimental procedures.

There have also been numerous papers questioning the health benefits of the many compounds that are in vitro antioxidants and whether they exhibit similar in vivo activity [19,20,21,22,23,24]. The U.S. Food and Drug Administration (FDA) and the European Food Safety Authority (EFSA) allow health claims for vitamin antioxidants (i.e., vitamins A, C, and E, those with a recommended daily intake (RDI) [25,26]. However, even for these compounds, there is some controversy with respect to their performance as antioxidants in vivo [22].

In the current review, we briefly describe oxidative stress, types of antioxidants, and in vitro assays, their mechanisms, their strengths and weaknesses, bioaccessibility, and bioavailability. We have also compared the analytical variations between the reported methodologies and activities for six commonly consumed food substrates (fruits, vegetables, processed foods, grains, legumes, milk, and dairy-related products). It is important for consumers, nutritionists, and other healthcare professionals to understand the health benefits gained from the consumption of fruits and vegetables and to distinguish facts from commercial hype regarding antioxidants.

## 2. Oxidative Stress

Oxidative stress is defined as the imbalance between the occurrence of reactive oxygen/nitrogen species (ROS/RNS) and cellular antioxidant defenses [1,4]. Oxidative stress is a result of excess ROS/RNS, which occurs due to a lack of counteraction by cellular antioxidant systems [1,5]. Increased oxidative stress can have severe consequences in biological systems, including molecular damage (such as nucleic acids, lipids, and proteins), which can severely impact health, as shown in Figure 1A [1,5]. Damage to biomolecules or the induction of several secondary reactive species due to oxidative stress ultimately leads to cell death (apoptosis or necrosis). It has been assessed that oxidative stress is associated with more than 100 diseases, including cardiovascular disease, cancer, hypertension, diabetes, neurogenerative diseases, aging, etc. [1,5].

Contrary to their harmful effects on health, ROS/RNS can have beneficial effects depending on their function, location, and amount. For instance, superoxide (O_2_^−•^) and nitric oxide (^•^NO) radicals at low or medium concentrations are involved in cellular responses and participate in signaling pathways [1]. H_2_O_2_, formed by various oxidase enzymes, and the action of superoxide dismutase (SOD), allows its use as an important signaling molecule, also it is substrate for generating further reactive species such as HOCl [27,28]. ROS are also involved in immunological responses, degrading xeno compounds and organisms through phagocytosis.

ROS are oxygen-containing molecules, including radicals (like the superoxide anion) and non-radicals (like H_2_O_2_) that greatly vary in their chemical abilities, such as diffusion in living cells and chemical reactivity with biomolecules. ROS examples include singlet oxygen, superoxide, hydrogen peroxide, and hydroxyl radicals (Figure 1B). Singlet oxygen is the highest energy spin state of molecular oxygen. In contrast to molecular oxygen in ground state, the two valence electrons are paired in an anti-bonding orbital. Singlet oxygen is therefore only generated, when molecular oxygen is energized via radiation. Importantly, and in contrast to other ROS subspecies, no electron transfer does occur during this process. Singlet oxygen is very reactive towards organic compounds and plays a deleterious role in biological systems, for instance, by involving in the oxidation of LDL cholesterol, which can lead to cardiovascular diseases. Moreover, increased ROS can trigger mtDNA mutations as well as promote uncontrolled proliferation and carcinogenesis [29]. The delicate balance of harmful and beneficial effects of free radicals is crucial for life processes, and antioxidants play an essential role in achieving this balance.

## 3. Antioxidants

Antioxidants can be broadly categorized in many different ways: (i) natural and synthetic; (ii) polar and non-polar; (iii) enzymatic and non-enzymatic; (iv) endogenous and exogenous; and (v) by the mechanisms in which they are involved [30]. Antioxidants primarily exhibit activities based on three mechanisms, hydrogen atom transfer, single electron transfer, and metal chelation [10]. They show their activity through three different pathways: (i) preventive: prevention of free radical formation and derivatives; (ii) interruption: interrupt radical oxidation reactions; and (iii) inactivation: inactivate free radical/radical derivative reaction products [30]. Endogenous antioxidants are primarily enzymes, such as superoxide dismutase (SOD), catalase (CAT), glutathione reductase (GR), and glutathione peroxidase (GPx). On the other hand, non-enzymatic endogenous antioxidants, such as glutathione and lipoic acid, are products of the body’s metabolism [2,31]. The first-line defense antioxidants (enzymatic) convert reactive superoxide and hydrogen peroxide into water and oxygen. The non-enzymatic antioxidants can act as a second-line defense against ROS by rapidly inactivating radicals and oxidants. The enzymatic antioxidants further act as the third-line defense involved in the detoxifying and removal. Dietary antioxidants, such as vitamins, carotenoids, polyphenols, flavonoids, and bioflavonoids (Figure 2), are exogenous antioxidants that have in vivo activity [30].

## 4. Antioxidant Assays

Various analytical methods have been developed to evaluate the antioxidant properties of plant-based phytochemicals [11,13,32,33]. The antioxidant activity depends on their chemical structure; specifically, it depends on their ability to donate hydrogen with electron, metal chelation, and their ability to delocalize the unpaired electron within the aromatic structure. Numerous analytical methods for evaluating each aspect of their antioxidant action, including either in vitro or in vivo, have been reported and discussed in the literature [34].

Antioxidant assays can be categorized into five mechanistic pathways, as summarized in Figure 3.

(i) Electron transfer-based assays: In these assays, a single electron transfer occurs between the antioxidant and substrate, which is measured to assess the potential of the plant’s secondary metabolites. Assays like cupric ion reducing antioxidant capacity (CUPRAC) [35,36,37], N,N-dimethyl-p-phenylenediamine dihydrochloride (DMPD) [38], ferric reducing-antioxidant power (FRAP) [39,40], Folin-Ciocalteu (FC), Trolox equivalent antioxidant capacity (TEAC) method/ABTS radical cation decolorization assay [41] come under this category.

(ii) Hydrogen atom transfer-based assays: In these assays, a hydrogen atom transfers from the antioxidant to the substrate. Assays such as oxygen radical absorbance capacity (ORAC) [42] and total radical-trapping antioxidant parameter (TRAP) [43] methods fall under this category.

(iii) Electron/hydrogen atom transfer (mixed) based assays: In these assays, the hydrogen atom transfer occurs via two-step mechanisms (Figure 3(iii)). Assays like DPPH scavenging activity [39,44,45] and TEAC follow this mechanism.

(iv) Metal chelation-based assays: In these assays, antioxidants chelate with transition metals like Fe(II) and Cu(II). Ferrous ion and cuprous ion chelating activity are examples of this category.

(v). Lipid oxidation and ROS/RNS scavenging activity assays: These assays are based on the ability of antioxidants in reducing/preventing lipid oxidation and scavenging ROS and RNS. β-carotene linoleic acid method/conjugated diene assay [45], ferric thiocyanate method (FTC) [39,46], thiobarbituric acid method (TBA), hydrogen peroxide (H_2_O_2_) scavenging assay [39,47], hydroxyl radical averting capacity method (HORAC) [42], nitric oxide scavenging activity [48], peroxynitrile radical scavenging activity, superoxide radical scavenging activity (SRSA/SOD), and xanthine oxidase methods come under this category.

(vi). Enzymatic antioxidant assays: Antioxidant enzyme systems that catalyze reactions to counterbalance free radicals and reactive oxygen species include superoxide dismutase and catalase. Catalase [49,50], ferric reducing ability of plasma [40,51,52], γ-glutamyl transpeptidase [53], glutathione peroxidase estimation [49,54], glutathione (reduced) GSH estimation [55], glutathione-S-transferase [56,57], LDL assay [58], lipid peroxidation assay [59], and superoxide dismutase method [60] can be categorized under enzymatic antioxidant assays.

Table 1 summarizes the various methods used to measure antioxidant activity. The principles, advantages, and limitations associated with each method, along with recent references, are also presented [13,14,18,36,46,51,61,62,63,64,65,66,67,68,69,70,71,72,73,74,75,76,77,78,79,80,81,82,83,84,85,86,87,88]. The advantages of electron transfer assays include faster reaction rates, ease of experimentation, sample throughput, and reproducibility. The main advantages associated with hydrogen atom transfer assays include close physiological resemblance, taking initiation and propagation into account, and uses of physiologically relevant radicals. Moreover, ORAC assay can be performed for antioxidants with a wide range of polarities, from lipophilic to hydrophilic [64]. Similarly, ROS/RNS scavenging activity/lipid oxidation assays, ET/HAT mixed, metal chelation, and lipid peroxidation inhibition assays have the advantages listed in Table 1.

The major limitation associated with all these assays is their lack of specificity. However, specific limitations include the solubility of antioxidants in the extraction solvent, interferences with coloring substances and other reducing phytochemicals, ignoring reaction kinetics, and not representing the physiological radicals used in these assays.

A second major limitation is the lack of equivalence of the methods. It is not possible to convert ORAC to FRAP values with a simple proportionality factor. As shown in the next section, some foods high in FRAP values may be low in ORAC values, and the opposite can be true. This situation is best summed up by stating that the FRAP assay generates FRAP values, ORAC generates ORAC values, DPPH generates DPPH values, etc., and there are no equivalency factors.

## 5. Antioxidant Activity of Selected Prominent Foods

We have summarized (Table 2) [89,90,91,92,93,94,95,96,97,98,99,100,101,102,103,104,105,106,107,108,109,110,111,112,113,114,115,116,117,118,119,120,121] peer-reviewed literature reports on the antioxidant activity/capacity data for six prominent groups of food and related products that are consumed globally: fruits (apples and berries), vegetables (spinach and olives), processed products (wine, coffee, and tea), dairy products (milk and yogurt), legumes (soybeans, beans), and grains (wheat and corn), documented by different researchers using various assay procedures. Results from each group are separately presented below. In each case, antioxidant assay methods are used to document changes in composition and emphasize the impact of genetics and processing on composition. However, it must be remembered that a specific antioxidant activity can be achieved in literally multiple ways by different possible combinations of components. Without data for specific components, it is impossible to relate antioxidant values to composition or health outcomes.

### 5.1. Fruits-Apples and Berries

Apples provide a rich source of phytochemicals, and epidemiological studies have shown that the consumption of apples reduces the risk of certain cancer types, cardiovascular diseases, asthma and diabetes. The antioxidant activity of apples has been attributed to various phytochemicals, particularly quercetin, catechin, phloridzin, and chlorogenic acid [122]. Antioxidant properties of different apple matrices (leaves, fresh fruit, pulp and peel, and pomace) have been investigated using various colorimetric assays (FC, DPPH, ABTS, and FRAP). The results from different matrices were expressed as gallic acid equivalents (GAE), ascorbic acid equivalents (AAE), Trolox equivalents (TE), and IC_50_ (50% inhibition concentration) (Table 2). For instance, Bahukhandi et al. [89] investigated the antioxidant activity of apples after pulverizing them to a fine texture. They studied the antioxidant activity of apples in terms of total phenolic contents using an FC reagent, and the results were expressed in GAE (10.87 mmol/kg). The antioxidant capacity for DPPH, ABTS, and FRAP was reported in AAE as 10.87, 24.57 mmol/kg, and 24.05 mmol/kg (of fresh apple), respectively. They also evaluated the correlation between total phenolic content, flavonoid, flavonol, TEAC_ABTS_, TEAC_FRAP_ and determined positive correlation values as 0.895, 0.843, 0.812, 0.856, and 0.830, respectively [89]. As noted, depending on the type of assay used, the reported values were significantly different. Additionally, significant variations in antioxidant activity were observed in different sample matrices (leaves, fresh fruit, pulp and peel, and pomace) even when the same assay was used. In a recent systematic review by Antonic et al. [123] on apple pomace, the authors showed that the high antioxidant content and dietary fibers present in apple pomace play an essential role as a food fortification ingredient in the food industry. The review highlights that fortified apple pomace increased the antioxidant activity and dietary fiber content in various food products. In a recent study, Li et al. [124] reported that red-fleshed apples showed greater antioxidant activities, phenolics, and flavonoid content than regular fuji apples. Particularly, one of the red-fleshed varieties, ‘A38’, showed about 3-fold higher FRAP, DPPH, and ABTS activities than the fuji apple [124]. The above results illustrate that proper documentation of genotypes is important. Unfortunately, none of these studies documented the difference in specific chemical components.

Berries, like apples, are considered to have several health benefits as they contain phenolic acids, flavonoids, and anthocyanins, which are localized mainly in berries, seeds, skins, and leaves [125]. For instance, blueberry anthocyanins are nature’s most potent antioxidants [125]. Similarly, blackberries show high antioxidant activity as they have highly abundant phenolic compounds, such as gallic acid, ellagic acid, ellagitannins, tannins, quercetin, cyanidins, and anthocyanins [126,127]. The antioxidant activities of berries are presented in Table 2. The data clearly shows significant variations in the results obtained from different assays. In another blueberry study, Liović et al. [128] studied the influence of freeze-drying, high-intensity ultra-sound, and pasteurization on gastrointestinal stability and antioxidant activity. The authors found that both total phenolic content and antioxidant capacities were improved for the freeze-dried and simulated gastric digested samples as compared to control untreated samples investigated in that study. These results suggested that external factors, such as drying and digestion, can also have a significant impact on bioactivity. Similarly, Dalmau and coworkers hypothesized that the drying process might alter the microstructures of vegetables as they found that both convective drying (CD) and freeze-drying (FD) decreased the TPC and antioxidant activity of beetroot samples [129]. The authors, however, claimed that these drying processes facilitate the better release of bioactive compounds during the digestion process and, in turn, lead to higher TPC and antioxidant activities [129].

### 5.2. Vegetables-Spinach and Olives

Spinach (*Spinacia oleracea*) is a vegetable with a wide array of phytoconstituents such as polyphenols, flavonoids, tocopherols, carotenoids, ascorbates, *p*-coumarins, vitamins, and polysaccharides, which are responsible for its nutritional properties [130,131]. The prominent antioxidants from spinach identified by different researchers include chlorogenic acid and spinacetins, and their analogs [132]. However, Mzoughi et al. reported the antioxidant activity of polysaccharides from spinach using DPPH, ABTS, and FRAP assays [90]. The water-soluble polysaccharides from *Spinacia oleracea* were extracted and characterized using FT-IR, UV–vis, ^1^H-NMR, SEC (Size Exclusion Chromatography)/MALS (multi-angle light scattering), and DRI (differential refractive index) techniques. The average molecular mass of the polysaccharide was 408 kDa composed of monosaccharides like arabinose, glucose, galactose, mannose, and rhamnose. *Spinacia polysaccharide* significantly prevented oxidation-induced Cd damage on HEK293 and HCT116 cells [90]. The results from both DPPH and ABTS assay were presented as percent inhibition (68.51 ± 0.89% and 70.12 ± 0.04%, respectively), whereas FRAP results were shown as reducing capacity in μmol/L (1590 ± 53.98 μmol/L at 10 mg/mL). On the other hand, Galla et al. [91] reported the antioxidant activity of the methanolic extract of spinach leaves assayed by DPPH, ABTS, and FRAP. In these two cases, one assay used water extraction to measure polysaccharides activity, and the other used methanol extraction to measure the activity of hydrophobic analytes. Hence, it becomes extremely challenging to identify the true antioxidant activity of foods using a single assay or extraction methodology. Another challenge involving these methods is the units used to report the activity. For instance, for the same assay (FRAP), Galla et al. [91] reported results in terms of percent inhibition, Hussain et al. [100] reported EC_50_, and Mzoughi et al. [90] reported reducing capacity in μmol/L [91]. Additional details on the antioxidant, antimicrobial activities, and clinical efficacy of *Spinacia oleracea* were presented in a recent review by Salehi et al. [131]. Recently, in 2020, Kamiloglu [133] reported the industrial freezing effects on the phenolic content and related antioxidant capacity of spinach. The results of both TPC and TAC (CUPRAC, ABTS, DPPH, and FRAP assays) showed that the freezing process increased both the TPC and TAC of spinach. Interestingly, undigested frozen samples have shown higher TPC and TAC results than digested (oral, gastric, and intestinal) samples [133]. These observations further illustrate that it is challenging to compare health claims just based on colorimetric assays commonly used for reporting antioxidant activities in foods.

Olives and olive oil have several health benefits due to the presence of phytochemicals [134,135]. Olives contain phenolic acids (caffeic acid, gallic acid) and their derivatives, phenolic alcohols (tyrosol, hydroxytyrosol), secoiridoids (oleuropein, oleocanthal), lignans (pinoresinol), and flavones (luteolin), which account for their antioxidant activity [134,135]. Hydroxytyrosol (HyT) is one of the main polyphenols found in virgin olive oil and olive mill waste [136] and has been shown to have strong ROS scavenging properties. HyT is the only phenolic compound that has received health claim approval from the European Food Safety Authority (EFSA). According to the EFSA, the consumption of olive oil polyphenols such as HyT and its derivatives play a protective role in preventing oxidative damage of blood lipids [136]. However, the antioxidant assays are not specific for HyT.

Recently, Fernández-Poyatos et al. [92] studied the antioxidant potential of table olives from *Olea europaea* L. They determined activity using conventional spectrophotometric methods ABTS and DPPH to be 308.68 μM and 228.46 μM Trolox equivalent per gram of dried extracts, respectively. Cheurfa et al. [105] investigated the antioxidant potential of the extract of olive leaves using water and aqueous ethanol separately. The ethanol extract of *O. europeae* leaves showed significant antioxidant activity (IC_50_ 69.15 mg/mL, % inhibition 54.98, % inhibition 49.71, 82.63 mg ascorbic acid equivalent/g extract, and 7.53 mol of Fe^2+^/g extract for the DPPH, β-carotene bleaching, ferric thiocyanate, TAC, and FRAP assays, respectively) [105]. As noted above, there are significant variations in the antioxidant capacity values for olives depending on the assay used and the units for the results.

### 5.3. Processed Products-Wine, Coffee, and Tea

Wine is primarily made from grapes but is also made from several other fruits, including apple, cherry, pear, peach, plum, banana, mango, strawberry, blueberry, blackberry, and raspberry [137]. These fruits are widely known for their antioxidant activity. Similarly, coffee and tea are also considered to have significant antioxidant activity [138]. The coffee-brewed drink is prepared from coffee beans, and tea is prepared from fresh tea leaf extracts. Antioxidant activities of these three processed food products (wine, coffee, and tea) have been widely studied using different assays, including FRAP, ABTS, DPPH, ORAC, and TPC [110,111,112,113,139,140,141,142]. However, the data in Table 2 showed significant variations in the outcomes from different assays, despite using the same sample for analysis [110,111,112,113,139,140,141,142]. Table 2 presents DPPH and ABTS studies by Jung et al. [110] reported in percent radical scavenging activity (RSA), whereas, for the same assays, Bravo et al. [111] reported results in μM TE/g. Moreover, the results from either study (DPPH vs. ABTS) were distinctly different, which can be attributed to the mechanisms involved in the assays. Similar variations were observed for tea and wine antioxidant activities (Table 2).

### 5.4. Legumes-Bean, Soybean

Soybean (*Glycine max* (L.) Merr.) is one of the most important plant protein sources consumed by humans and animals and receives growing interest as a source of high-protein crops in Europe, North America, South Asia, and Japan [115]. The antioxidant activities of different matrices of soybeans have been studied, as shown in Table 2. However, for each assay, different units were reported, which makes it difficult to compare the antioxidant potential of different foods. For instance, Peiretti et al. [115] reported the TPC activity in catechin eq/g, whereas Handa et al. [93] reported in GA eq/g.

Common beans (*Phaseolus vulgaris* L.) are one of the most important legumes consumed globally and are used for nutritional and medical purposes [125]. Beans are rich in polyphenols, flavonoids, anthocyanidins, and procyanidins, which may explain their antioxidant activities. Zhao et al. [101] and Cid-Gallegos et al. [114] reported antioxidant activities for chickpeas. The results presented by the authors are not comparable as they used different solvents for extractions and expressed their results in different units (Table 2).

### 5.5. Grains-Corn, Wheat

Epidemiological studies have shown that the consumption of whole grains and grain-based products is associated with a reduced risk of oxidative stress-related chronic diseases and age-related disorders, such as cardiovascular diseases, carcinogenesis, type II diabetes, and obesity [143]. The health benefits of whole-grain flours are attributed to the presence of antioxidants, such as tocopherols and tocotrienols, vitamin E, carotenoids, phenolic acids, and flavonoids [144]. In 2019, Ranjbar et al. [117] studied the antioxidant potential of wheat flour with iron enrichment using DPPH and FRAP assays. However, the authors reported phenolic acid content as μg equivalent of ferulic acid/g, whereas, usually, the TAC is reported as either gallic acid or ascorbic acid equivalents/g. Hence, the comparison of antioxidant results becomes complicated for researchers and consumers.

Corn is among the largest produced staple foods across the globe. Corn is well known for its antioxidant properties, which may be attributed to the presence of high amounts of anthocyanins [145]. Antioxidant studies by Ramos-Escudero et al. [94] and Horvat et al. [95] showed no significant relationship between TPC and DPPH radical scavenging in corn samples (r = 0.202) [95]. This lack of correlation may be due to several reasons, such as experimental conditions, mechanisms, interferences, and types of the analytes assayed [95].

### 5.6. Dairy Products-Milk, Yogurt, and Others

Dairy products constitute about 25–30% of the average diet of an individual [146]. Milk and milk products are rich in essential nutrients such as vitamins (tocopherol, retinol, and carotenoids), minerals, oleic acid, omega-3 fatty acids, conjugated linoleic acid, antioxidants like milk caseins, ascorbate, low molecular weight thiols, and whey proteins, and other bioactive compounds [146,147].

Zulueta et al. [120] reported the antioxidant capacity of multiple commercial samples of pasteurized and ultra-high temperature (UHT) treated whole milk, whey, and deproteinized milk using ORAC_FL_ assay [120]. According to the study, the TAC of whole milk was attributed mainly to casein fractions, albumin, and whey protein, whereas hydrophilic antioxidant compounds, such as ascorbic acid and uric acid, were the main contributors to the total TAC of the deproteinized milk. A significant correlation was found between the fat% and the TAC of milk samples. In addition, pasteurized milk was found to have significantly higher TAC than UHT-treated milk for both whey and deproteinized milk samples. In contrast, the TAC values of pasteurized and UHT whole milk were not significantly different [120]. The reported antioxidant results were entirely based on a single ORAC_FL_ assay, which is the main limitation of this report, as no single assay is expected to provide an accurate measurement of total antioxidant capacity. In addition, this report also suggested that sample processing plays a very important role in the overall antioxidant activity/capacity of the food substrate.

In a recent study by de Carvalho et al. [121], yogurt samples were fortified with 0.25% and 0.5% freeze-dried stevia extract (FSE). The control and stevia-fortified yogurts were evaluated and compared for the TPC and antioxidant activity (using FRAP and ABTS). Table 2 shows the TPC, FRAP, and ABTS results for the yoghurt 0.14 ± 0.01 mg GAE/g, 0.40 ± 0.03 μmol Trolox/g, 0.40 ± 0.04 μmol Trolox/g, respectively. Upon adding 0.25% FSE, the antioxidant activity of the yoghurt significantly increased (TPC- 0.43 ± 0.02 mg GAE/g, FRAP- 2.57 ± 0.09 μmol Trolox/g, ABTS- 3.63 ± 0.08 μmol Trolox/g. Moreover, upon addition of 0.5% FSE, the antioxidant activity further increased TPC- 0.65 ± 0.02, FRAP- 4.19 ± 0.05 μmol Trolox/g, ABTS- 5.34 ± 0.23 μmol Trolox/g. Apart from FSE, the simulated digestion also increased the antioxidant activity of the fortified yogurts compared to the undigested fractions [121].

## 6. Strengths and Weaknesses of Antioxidant Assays

### 6.1. Strengths

In general, antioxidant assays provide rapid and inexpensive means of measuring the status of a sample. Changes in the composition of a food or supplement arising from genetics, environment, management, and processing are readily detected. Antioxidant assays are sensitive to the metadata associated with a sample. The assays cannot specify which components are changing, but changes are readily detected.

### 6.2. Weaknesses

Questions have been raised about the relationship between dietary/exogenous antioxidants and health. Lack of specificity, lack of harmonization, and the inability to correlate in vitro measurements with in vivo activity have been major factors. The U.S. Food and Drug Administration (FDA) and the European Food Safety Authority (EFSA) do allow health claims for vitamins (A, C, and E), which have shown ambiguous results in vivo. Several journals have banned papers whose primary measurements are antioxidant activity. This suggests the complexity of the issue [148]. The major limitations of currently used antioxidant assays are summarized below.

(a) There is a huge divergence in the results for food and dietary antioxidants because the assays are non-specific and are influenced by every component in a sample matrix. A change in an assay value cannot be correlated with a change in a specific component. Consequently, antioxidant assays cannot be correlated with health outcomes.

(b) Antioxidant assays cannot be compared or inter-converted. Antioxidant assays measure activities. A detailed explanation for these limitations has been described by Apak et al. [15]. Both AOAC International (Association of Analytical Communities) and the IUPAC (International Union of Pure and Applied Chemistry) reported that the results from different antioxidant methods could not be compared as each method employs different mechanisms, pH, temperature, and sample matrix [148]. Hence, the IUPAC concluded that there is no single universally accepted antioxidant assay available that accurately determines the antioxidant activity or the total antioxidant capacity [10,60,148]. In 2012, the U.S. Department of Agriculture (USDA) removed the ORAC database from online ‘because of increasing evidence that the data infers there is no relevance between antioxidant activity and the effects of bioactive compounds, including polyphenols on human health’ [10,148].

(c) Reporting results in multiple units for the same assay complicates data comparison [149]. For example, results for TPC have been reported as equivalents of gallic acid, ascorbic acid, or ferulic acid, which complicates the understanding of the results.

(d) The measurements in vitro experiments cannot be directly correlated with in vivo activity. This can be attributed to the complex physiological environment in vivo assays as compared with the controlled environments in vitro assays [149,150]. The activity of plant-based secondary metabolites such as polyphenols, phenolic acids, flavonoids, and proanthocyanidins, is little known [148].

(e) The interpretation of antioxidant assay results mainly focuses on the antioxidant-oxidant reactions and related kinetics; however, they often ignore the chromophore/luminescent interactions with the probe, which can cause interferences in the results [151].

## 7. Other Factors Influencing Antioxidant Activity

Unlike the in vitro assays, the measurements based on in vivo antioxidant assays are impacted by several factors relevant to physiological conditions. For instance, the fate of the antioxidant in vivo is determined by the pharmacokinetic phenomena ADME (absorption, distribution, metabolism, excretion) profiles, which are not evaluated in vitro [152]. Unlike in vitro, the actual concentrations of antioxidants at tissue levels are essentially depending on ADME profiles. Below are some of the critical limitations of in vivo antioxidant assays [15,152,153].

### 7.1. Bioaccessibility and Bioavailability of Antioxidants

Bioaccessibility and bioavailability are closely related, but they have different definitions. Bioaccessibility is the fraction of bioactive compounds that are released from a matrix during the digestion process and become available for absorption, whereas bioavailability is the fraction of compounds that are absorbed into the bloodstream, distributed by systemic circulation, and exert their effect after being metabolized then eliminated [154,155]. For phytochemicals, as with any food component, to exert their biological activity, they must be released from the matrix in an absorbable form into the stomach/intestine/colon (bioaccessible), followed by absorption into the bloodstream (bioavailable) [154,155]. Factors such as matrix interactions and chemical structures influence the bioaccessibility of phytochemicals, whereas bioavailability can be affected by multiple parameters, such as biological membranes (GI wall), the physicochemical properties of the phytochemicals, biological environment inside the GI, etc., The optimum properties of phytochemicals/drugs for GI absorption have been defined by Lipinski’s Rule [156], including appropriate molecular weight, hydrogen bonding capability, and partition coefficient (LogP) [156]. In addition, the presence of adjuvants, food processing techniques, sample preparation, and extraction methods can also significantly influence the bioavailability of the compounds by influencing their bioaccessibility [154,155].

The bioaccessibility and bioavailability of phytochemicals can be enhanced using processing techniques that induce physical or chemical modifications in the food [154]. These modifications include: (a) chemical modifications, such as cleavage of hydrophobic forces, hydrogen bonds, and bonds that attach phenolic compounds to matrix macromolecules, conjugation, and derivatization; (b) physical modifications, such as grinding, drying by different approaches; (c) disrupting the cell wall barriers so that phytochemicals can be released from the matrix; and (d) using encapsulation techniques/solubilization using nanotechnologies that protect phytochemicals until they are absorbed. One should note that these techniques can cause the degradation of phytochemicals; however, it is possible to reduce it by altering the operating conditions, which can facilitate increased bioaccessibility and bioavailability [154]. Techniques such as drying, freezing, thermal processing, sterilization and pasteurization, ultrasounds treatment, milling and grinding, chemical and enzymatic treatments, and encapsulation, which are commonly used for food processing and supplements, can influence the overall phytochemical availability [154]. The positive or adverse effects of processing techniques on bioaccessibility and bioavailability depend on compounds’ stability, matrix protective effects, and existing interactions.

Bioaccessibility, bioavailability, the metabolism of antioxidants, and their consequences, must be taken into consideration. For instance, flavonoids are structurally altered in vivo. Hence, the nutritional application of flavonoids requires extensive studies on their metabolism and controlled comparison of antioxidant activity of their structural isoforms. Additionally, the evidence shows that the removal of glycosidic substituents (sugar moiety) by enzymes or bacteria is likely to increase the antioxidant activity of flavonoids in vivo. In contrast, the methylation and glycosylation of OH groups in flavonoids reduce their prooxidant behavior by catechol-o-methyltransferase (COMT) and other enzymes. The impact of sugar moiety on the bioactivities of flavonoids has been discussed in recent reviews by different researchers [153,157,158].

### 7.2. Chelation

The efficacy of antioxidants’ function based on metal chelation mechanism, such as polyphenols, need to be investigated thoroughly in vivo. Though ascorbate is an antioxidant, it can also act as a pro-oxidant, especially in the presence of transition metals like iron and copper [159]. Hence, it is important to understand the actual factors contributing to the antioxidant activity in the presence of metals, ascorbate, and other possible interferences, such as uric acid, which is elevated upon consumption of polyphenol-rich foods and may cause increased plasma total antioxidant capacity in vivo. Finally, it has been suggested that the large increase in plasma total antioxidant capacity observed after the consumption of polyphenol-rich foods is not caused by the polyphenols themselves but is likely the consequence of increased uric acid levels [160].

### 7.3. In Vivo Assays

Over the years, researchers have discussed the validity of applying various assays to in vivo conditions. All of them provide an estimate of the antioxidant capacity of plasma/serum without distinguishing the contribution by exogenous molecules of dietary origin (i.e., ascorbic acid, vitamin E, polyphenols, and other phytochemicals) compared to endogenously-derived molecules such as enzymatic components (i.e., SOD, GSH-Px, and catalase, CAT) and small macromolecules (i.e., albumin, bilirubin, ceruloplasmin, ferritin, glutathione) [161]. In this regard, the EFSA remarked on the inappropriateness of the methods used to determine antioxidant capacity in humans and extrapolate possible effects on human health [162].

### 7.4. Sample Matrix

Sample/substrate matrix always has a great influence on the outcomes of antioxidant studies, both in vitro and in vivo. Along with matrix effects and multiple variable experimental conditions, performing the assays (radical generator, time of measure, expression of results, etc.) makes it difficult to compare the reported data.

### 7.5. Experimental Parameters

Most assays lack a detailed evaluation of important factors that affect the antioxidant activity measurements, such as concentration, pH, localization, distribution, metabolism, reactivity toward other non-targeted molecules, interaction with other antioxidants, and the fate of the radical that is derived from the antioxidant.

## 8. Perspectives/Recommendations

To achieve absolute efficacy of food antioxidants acceptable to various communities such as nutrition and clinical professionals and consumers, it is essential to consider the following factors:

(a) Matrix information must be presented in detail in the experimental section of the in vitro and in vivo antioxidant assays. The matrix effects should also be accounted for.

(b) Sample preparation (grinding, drying, processing, etc.) and the extraction procedures for antioxidants have a direct impact on the results. Hence, one should use optimized sample preparation and extraction methods to extract antioxidants with a wide range of polarities.

(c) Since there is no universal standard available currently, the development of universal multi-component standards is essential to address the variances associated with antioxidant capacity measurements.

(d) Both antioxidant activity and antioxidant capacity must be distinguished as these two terms are used interchangeably even though they are both measured completely differently, i.e., antioxidant activity is measured kinetically, and antioxidant capacity is measured thermodynamically. An ideal method for an antioxidant activity measurement requires: (i) usage of a biologically relevant radical source; (ii) determines actual chemistry that occurs in the assays; (iii) the use of a method with a defined endpoint and chemical mechanism involved in a particular assay; (iv) a suitable method for both hydrophilic and lipophilic antioxidants; and (v) instrumentation, which is readily available, simple, economical, rugged, user friendly, and facilitates high-throughput for routine analyses.

(e) The harmonization of methodology is required, which includes detailed standard operating procedures for different assays, test conditions, good internal/external standards, proper method validation, including intra- and inter-lab validation (reproducibility, accuracy, precision, and recovery), quality control, and quality assurance.

(f) All antioxidant results should be presented in a single international system of units (SI). Correlations tables between various assays should be established to promote easy comparison of results between different assay procedures.

(g) A better understanding of the bioaccessibility and bioavailability of antioxidants in vivo is essential. To achieve maximum efficacy from the antioxidants in vivo, their bioaccessibility and bioavailability must be increased by using appropriate delivery systems, such as liposomes, nanoparticles, etc.

Overall, it is not easy to accomplish the above recommendations altogether. However, focusing on each of the above steps will improve the credibility of antioxidants. This will enable nutrition professionals, and researchers to correlate accurately the role of antioxidants as it relates to nutrition and health.

## 9. Conclusions

The present review defines antioxidants and describes antioxidant assays, their mechanisms, and their strengths and weaknesses. These methods have been used to evaluate the “health benefits” of phytochemicals, as documented in the literature. Unfortunately, with antioxidant assays, it is not certain what has been measured. The lack of specificity of the assays creates confusion and ambiguity for consumers, healthcare professionals, and researchers. To have a wide acceptance and clear understanding of the health benefits of phytochemicals, there is a critical need to develop multi-omic approaches which measure specific food and supplement components. This will allow health outcomes to be correlated with specific food components.

## Figures and Tables

**Figure 1 antioxidants-11-02388-f001:**
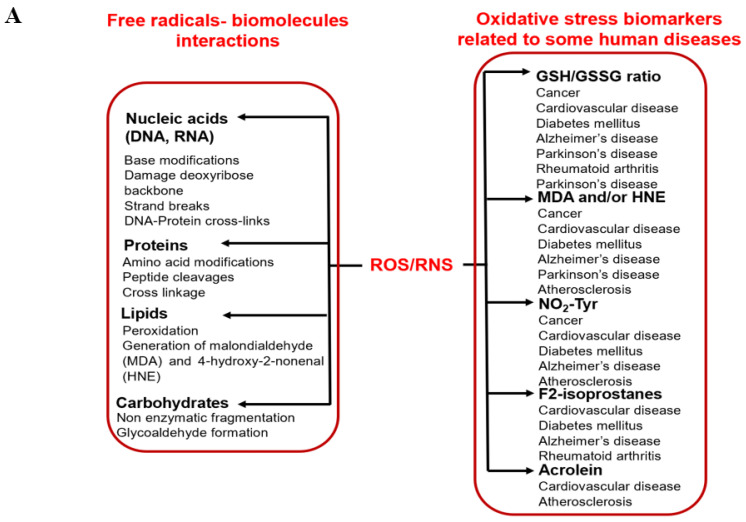
Schematic representation of (**A**) radicals’ impact on human health, and (**B**) the generation of various radicals in vivo.

**Figure 2 antioxidants-11-02388-f002:**
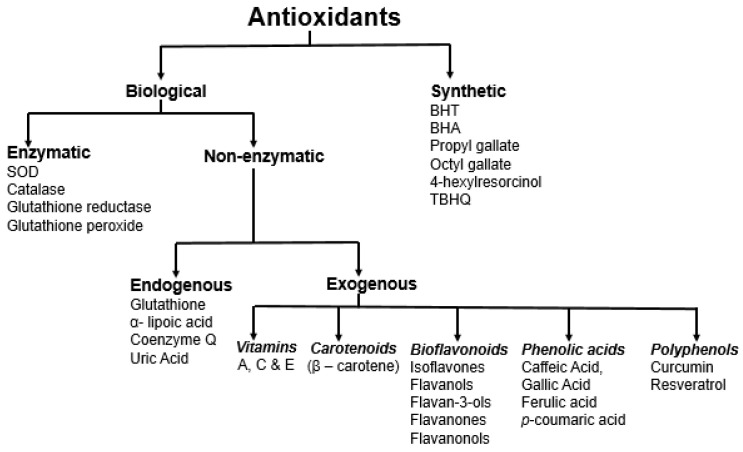
Different approaches for the classification of antioxidants.

**Figure 3 antioxidants-11-02388-f003:**
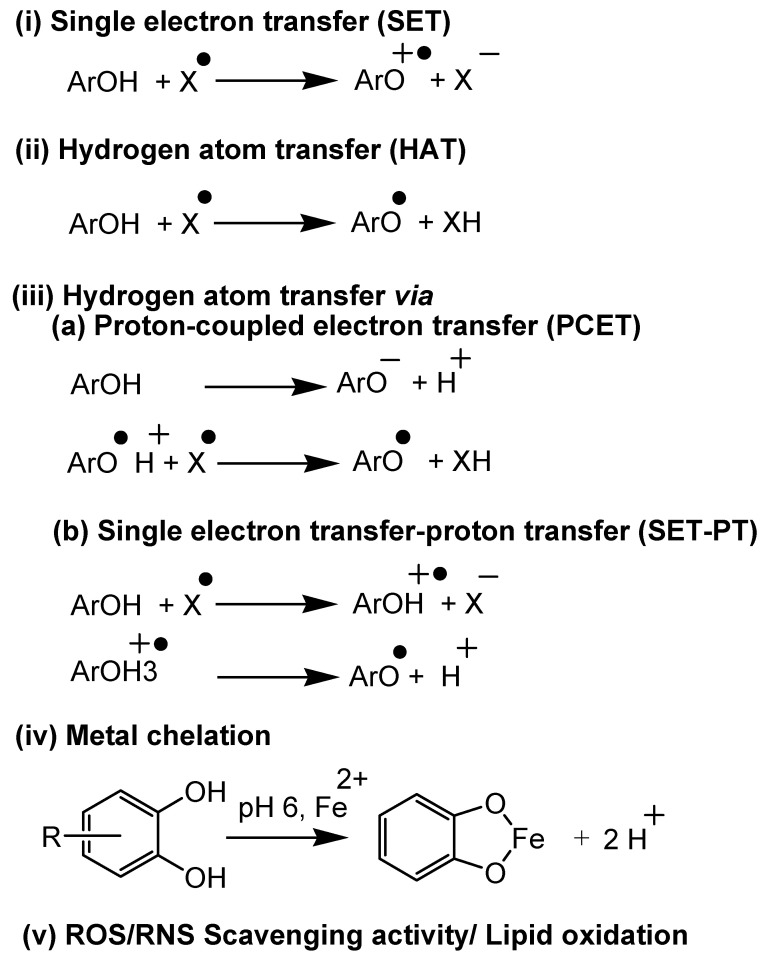
Mechanistic aspects of antioxidant assays both in vitro and in vivo.

**Table 1 antioxidants-11-02388-t001:** Various in vitro and in vivo antioxidant assays, their principles, advantages, and limitations.

Mechanism (Category)	Assay	Technique/Principle	Advantages	Limitations	Ref.
	In vitro				
Electron transfer (Total Antioxidant Capacity)	CUPRAC (Cupric ion reducing antioxidant capacity method)	In this assay, phenolic groups in the polyphenols are oxidized to quinones, whereas Cu(II) is reduced to Cu(I), which is measured at 450 nm.	(+) Copper reaction rates are faster than that of ferric ions, and it is more specific for antioxidants	(−) it ignores the reaction kinetics	[36]
	DMPD (N,N-dimethyl-p-phenylene diamine dihydrochloride) method	In the presence TROLOX, reduction of DMPD radical cation by antioxidants, the absorbance at 505 nm is decreased.	(+) easy, cheaper, and reproducible	(−) it ignores the reaction kinetics, and the DMPD radical is a non-physiological radical	[62]
	FRAP (Ferric reducing-antioxidant power assay)	Antioxidants at low pH reduce ferric-tripyridyltriazine (Fe^III^-TPTZ)to Fe^II^ form which is measured at 593 nm.	(+) a good representation of electron transfer mechanism (+) the is inexpensive, easy to prepare reagents, reproducibility, and speedy and a straight forward procedure	(−) it ignores the reaction kinetics and non-specific to antioxidants	[51]
	Follin-Ciocalteu reducing capacity	In this assay, phenols are oxidized in a basic medium by a mixture of tungstate and molybdate (Folin-Ciocalteu reagent), with the consequent formation of colored molybdenum ions, MoO_4_^+^ (750 nm).	(+) easy and reproducible	(−) it ignores the reaction kinetics and non-specific to antioxidants	[14]
	Trolox equivalent antioxidant capacity (TEAC) method (ABTS radical cation decolorization assay)	Upon reaction with an antioxidant (Trolox), ABTS (2,2-azo-bis(3-ethylbenz-thiozoline-6-sulfonic acid)) radical cation, which is a blue-green chromophore, reduces and decolorized. This assay uses a diode-array spectrophotometer at 750 nm.	(+) it can screen both hydrophilic and lipophilic antioxidants, easy and reproducible	(−) it ignores the reaction kinetics, and the ABTS is a non-physiological radical (−) The assay is not suitable for the determination of proteins antioxidant activity	[63]
Hydrogen Atom Transfer (Antioxidant Activity)	ORAC (Oxygen radical absorbance capacity) method	AAPH (2,2-azobis-2-aminopropane dihydrochloride) decomposition induces peroxyl radicals, and radical scavengers are used to measure the decrease in fluorescence. AAPH is used as a radical generator and Trolox as the antioxidant control. 485 nm is used as the excitation wavelength, and 520 nm is used as the emission wavelength.	(+) physiologically resemble method, and it takes initiation and propagation into account	(−) lack of consistency and the possible underestimation of antioxidant activity as B-PE can interact with phenolic acids (−) The method has been reported to fail determining both hydrophilic and lipophilic antioxidants	[64]
	TRAP (Total radical-trapping antioxidant parameter method), both in vivo and in vitro	In this method, the antioxidant potential is assessed by measuring the decay in decoloration. ABAP (2,2′-azo-bis(2-ami-dino-propane)hydrochloride) is a radical initiator that quenches the fluorescence of R-Phycoerythrin (R-PE).	(+) peroxyl radical is a common and physiologically representative radical	(−) detection probe(oxygen) that is not stable and may cause issues in measurements	[65,67,68]
ROS/RNS scavenging activity/lipid oxidation	β-carotene linoleic acid method/conjugated diene assay	The ROS oxidizes linoleic acid, and the resulting products initiate β-carotene oxidation, which leads to discoloration. In the presence of antioxidants, the discoloration will be delayed and measured at 434 nm.	(+) shows a strong correlation with the total phenolics measured by the F-C method.	(−) lack of reproducibility and crude kinetic treatment	[14,68,69]
	Ferric thiocyanate (FTC) method	During linoleic acid peroxidation, peroxides were formed, which oxidize Fe(II) to Fe(III). The Fe(III) reacts with thiocyanate to form a red color complex, which is measured at 500 nm.	(+) used to measure peroxide amount at the starting phase of peroxidation	(−) lack of specificity	[46,70,71]
	Thiobarbituric acid (TBA) method	In this assay, TBA and trichloroacetic acid are mixed with the sample solution, placed in the hot water bath for 10 min, centrifuged in the solution, and supernatant absorbance activity is measured at 552 nm.	(+) used to measure the concentration of free radicals present at the end of peroxide oxidation	(−) Not specific	[14]
	Hydrogen peroxide scavenging (H_2_O_2_) assay	Antioxidants reduce hydrogen peroxide concentration, which is measured at 230 nm using a spectrophotometer.		(−) most plant and food samples also absorb at this wavelength, which can compromise both the precision and accuracy of the method	[72]
	Hydroxyl radical scavenging activity	In the presence of antioxidant, the degraded product of deoxyribose (TBARS) measured colorimetrically at 532 nm.	(+) Useful for ketone containing antioxidants	(−) Higher concentration of antioxidants required	[73]
	Nitric oxide scavenging activity	Under aerobic conditions, nitric oxide reacts with oxygen to form nitrate and nitrite, which can be quantified using Griess reagent, and the absorbance is measured at 546 nm.	(+) relatively simple experimentation and physiologically relevant	(−) detection technique is not easily available and has a long reaction time	[18,74]
	Peroxynitrile radical scavenging activity	ONOO.scavenging activity is measured by the oxidation of dihydroxyrhodamine to rhodamine fluorescence spectrophotometer with an excitation wavelength of 485 nm and emission wavelength of 530 nm.	(+) Peroxynitrile is a good oxidizing agent for dihydroxyrhodamine	(−) Under anaerobic conditions, nitric oxide did not oxidize dihydrorhodamine and inhibited spontaneous oxidation of dihydrorhodamine	[75]
	Superoxide radical scavenging activity (SRSA/SOD)	This assay is based on the removal rate of superoxide radical (O_2_^−^) using antioxidants, which is measured by nitro blue tetrazolium (NBT) at 560 nm.	(+) peroxyl radical is a common and physiologically representative radical	(−) irreproducibility due to the water insolubility issue of diformazan, the end product of NBT reduction	[76,77]
	Xanthine oxidase inhibition assay	Xanthine is a substrate in XOD- catalyzed reaction, which yields uric acid as a product. Allopurinol is used as a xanthine oxidase inhibitor, measured at 293 nm.	(+) Possible to get kinetics	(−) Enzyme collection is tricky	[8,78]
ET/HAT_mixed	DPPH scavenging activity	1-diphenyl-2-picrylhydrazyl (α,α-diphenyl-β-picrylhydrazyl; DPPH) is a stable free radical due to electron delocalization, which prevents its dimerization. DPPH reacts with antioxidants, which diminishes its deep violet color, which is measured at 517 nm (515–518 nm).	(+) easy and reproducible	(−), difficult to get the reaction kinetics, and the DPPH radical is a non-physiological radical	[66]
Metal chelation	Ferrous ion chelating activity assay/Ferrozine assay-Fe(II)	TAC assay obtained via reduction of Fe(III) to Fe(II), and formed Fe(II) is determined with ferrozine using spectrophotometric absorbance measurement at 562 nm.	(+) High sensitivity, correlated with structure-activity relationships, higher molar absorptivity, relatively lower interference from foreign ions, wide pH tolerance, complex stability constant, water solubility, and low viscosity	(−) Not correlated with FRAP, DPPH, and TPC	[87,88]
	Cuprous ion chelating activity/Pyrocatechol violet-Cu(II)	Free Cu(II) that is not complexed with antioxidants is bound to Pyrocatechol, which is assessed at 632 nm.	(+) good repeatability and reproducibility, Cu^2+^ chelating ability is significantly and positively correlated to DPPH, FRAP, and total phenolic content		[88]
	In vivo				
Hydrogen atom transfer	Catalase (CAT)	The catalase activity is measured in an erythrocyte lysate as the difference in absorbance (λ_240_) per unit as the H_2_O_2_ maximum absorption wavelength is 240 nm. Catalase activity is used both in vivo and in vitro.	(+) a good representation of physiological conditions		[78]
Electron transfer/reducing power (Total Antioxidant Capacity)	Ferric reducing ability of plasma (FRAP)	This assay is primarily based on the principle that, at low pH, ferric-tripyridyltriazine (Fe^III^-TPTZ) is reduced to Fe(II). The antioxidant capacity is measured using the increased Fe^II^, which is measured spectrophotometrically at 593 nm.	(+) most simple, rapid, inexpensive tests and very useful for routine analysis, a good representation of electron transfer mechanism	(−) it ignores the reaction kinetics and non-specific to antioxidants	[51,79]
	γ-glutamyl transpeptidase (GGT)	GGT transfers the γ-glutamyl group from the L-γ-Glutamyl-p-nitroanilide and liberates the chromogen p-nitroanilide (pNA, 418 nm) proportional to the GGT present.	(+) a good representation of physiological conditions		[80]
Lipid peroxidation inhibition	Glutathione peroxidase (GSHPx) estimation	GSHPx is a seleno-enzyme that catalyzes the reaction of hydroperoxides with GSH to form GSSG and reduction of hydrogen peroxide.	(+) a good representation of physiological conditions		[13,85]
	Glutathione reductase (GR) assay	GR catalyzes the reduction of GSSG to GSH. GR activity is determined at 340 nm and 412 nm. One may expect a decrease of activity at 340 nm as a result of the oxidation of NADPH or an increase at 412 nm caused by the reduction of dithiobis (2-nitrobenzoic acid) DTNB.	(+) a good representation of physiological conditions		[13,81]
	Glutathion-S-transferase (GSt)	This assay utilizes 1-Chloro-2,4-dinitrobenzene (CDNB). Potassium phosphate, GSt, and CDNB mixture are incubated at 37 C, pH 6.5 for 5 min, followed by adding substrate. 340 nm absorbance is used for monitoring the assay.	(+) a good representation of physiological conditions		[13]
	LDL assay	The extent of low-density lipoprotein (LDL) oxidation is determined by the amount of lipid peroxides, also by using a thiobarbituric acid reactive substances (TBARS) assay determined at 532 nm.	(+) LDL is a true representation of physiologically	(−) limitations in the isolation of LDL from the blood, and it is difficult to monitor the lag phase	[13,83,84,85]
Lipid peroxidation inhibition	Lipid peroxidation (LPO) assay	Malondialdehyde (MDA) is one of the end products of lipid peroxidation, which is used for the LPO assay measured at 586 nm.	(+) a good representation of physiological conditions		[84,85]
	Superoxide dismutase (SOD) method	The SOD assay works based on the absorbance change at 420 nm related to pyrogallol.	(+) a good representation of physiological conditions		[86]

**Table 2 antioxidants-11-02388-t002:** Variations in the in vitro antioxidant activity for six prominent groups of food and related products that are consumed globally: fruits (apple and berries), vegetables (spinach and olives), processed products (wine, coffee, and tea), dairy products (milk and yogurt), legumes (soybeans, beans), and grains (wheat and corn), as documented in peer-reviewed published literature.

Samples	Matrix	Assay	Results	Ref.
** *Fruits* ** **Apple**	Fresh apple	TPC	6.82 mg GAE/g fw for Benoni cultivars from the location Mukhwa	[89]
		DPPH	10.87 mmol AAE/kg fw	
		ABTS	24.57 mmol AAE/kg fw	
		FRAP	24.05 mmol AAE/kg fw	
	Fresh apple	TPC	4.18 ± 0.1 mg GAE/g dw	[96]
		DPPH	22.14 ± 1.2 μmol TE/g dw	
		FRAP	26.98 ± 0.9 μmol TE/g dw	
		ABTS	32.85 ± 1.5 μmol TE/g dw	
	Apple peel	TPC	0.48 g GAE/kg	[102]
		DPPH	121 mol TEAC/kg	
		ABTS	13 mol TEAC/kg	
	Wild apples peel and pulp (ultra-sonic extract)	TPC	8.00 mg GAE/g fw in peel	[97]
			6.64 mg GAE/g fw in pulp	
		DPPH	IC_50_: 240.00 ± 6.00 μg/mL peel	
			IC_50_: 286.00 ± 7.00 μg/mL pulp	
		ABTS	IC_50_: 134.00 ± 3.00 μg/mL peel	
			IC_50_: 167.00 ± 4.00 μg/mL pulp	
	Apple pomace	TPC	3.48 ± 0.12 mg GAE/g apple pomace for MeOH extract	[98]
		DPPH	72.6 ± 1.6% (Inhibition)	
		FRAP	65.8 ± 1.8% (Inhibition)	
		ABTS	84.3 ± 1.6% (Inhibition)	
	Apple leaves	TPC	143.84 ± 37.79 mg GAE/g	[99]
		DPPH	259.68 ± 46.91 μmol TE/g	
		ABTS	625.26 ± 141.31 μmol TE/g	
		FRAP	328.02 ± 130.38 μmol TE/g	
**Berries**	Blueberry	TPC	443.60 ± 17.00 mg GAE/g	[100]
		DPPH	87.90 ± 0.20% inhibition (100 μg/mL); IC_50_ 1.40 ± 0.10 μg/mL	
		ABTS	23.10 ± 0.60% inhibition (100 μg/mL); IC_50_ 14.00 ± 0.50 μg/mL	
	Blackberry	TPC	269.5 ± 16 mg GAE/g	
		DPPH	77.80 ± 2.00% inhibition (100 μg/mL); IC_50_ 1.30 ± 0.10 μg/mL	
		ABTS	25.30 ± 1.10% inhibition (100 μg/mL); IC_50_ 23.00 ± 5.00 μg/mL	
	Black raspberry	TPC	965.60 ± 2.90 mg GAE/g	
		DPPH	89.03 ± 0.040% inhibition (100 μg/mL); IC_50_ 3.40 ± 0.40 μg/mL	
		ABTS	21.3 ± 1% (per 100 μg/mL); IC_50_ 79.00 ± 18.07 μg/mL	
	Red raspberry	TPC	434.3 ± 6.3 mg GAE g^−1^	
		DPPH	87 ± 1.2% inhibition (100 μg/mL); IC_50_ 1.40 ± 0.10 μg/mL	
		ABTS	31.1 ± 0.6% inhibition (100 μg/mL); IC_50_ 15.00 ± 0.90 μg/mL	
	Strawberry	TPC	250.10 ± 17.10 mg GAE/g	
		DPPH	70.20 ± 1.00% inhibition (100 μg/mL); IC_50_ 3.1 ± 0.02 μg/mL	
		ABTS	26.20 ± 0.70% inhibition (100 μg/mL); IC_50_ 9.9 ± 0.40 μg/mL	
	Lowbush blueberry	TPC	24.50 ± 0.69 mg GAE/g	[103]
		ABTS	127.00 ± 5.30 μmol TE/g	
		FRAP	389.00 ± 19.40 μmol FeSO_4_ equivalent/g	
** *Vegetables* ** **Spinach**	Dried, powdered	DPPH	36.71% inhibition (180 μg sample/mL)	[91]
		ABTS	68.34% inhibition (180 μg sample/mL)	
		FRAP	0.14% inhibition (180 μg sample/mL)	
	Gamma irradiated (above 0.75 kGy) samples	DPPH	EC_50_ 42–50% inhibition	[104]
		FRAP	EC_50_ 0.48–0.7% inhibition	
		TPC	208.9–216.2 mg GAE/g	
	Polysaccharides	DPPH	68.51 ± 0.89% inhibition	[90]
		ABTS	70.12 ± 0.04% inhibition	
		FRAP	1590 ± 53.98 μmol/L at 10 mg/mL BHT and AA	
**Olives**	Lyophilized table Olive; methanol extract	TPC	31.52 mg GAE/g	[92]
		ABTS	308.68 μmol TE/g	
		DPPH	228.46 μmol TE/g	
	Leaves; ethanol extract	DPPH	69.15 ± 0.06% Inh	[105]
		β-carotene bleaching	54.98 ± 0.03%	
		TPC	82.63 ± 0.02 mg AAE/g extract	
		FRAP	07.53 ± 0.06 mol Fe^2+^/g extract	
	Sprouted olive seeds	TPC	~4.50 mg GAE/g dw	[106]
		ABTS	~12 μmol TE/g dw	
		DPPH	~11 μmol TE/g dw	
		FRAP	~9 μmol TE/g dw	
** *Processed food* ** **Wine**	Red wine	TPC	317.62 ± 18.75 mg/mL	[107]
		DPPH	3.16 ± 0.15 mg GAE/mL	
		ABTS	7.10 ± 0.75 mg TE/mL	
		FRAP	8.20 ± 0.76 mg TE/mL	
	Standard white wine	FRAP	336.70 ± 15.20 μmol TE	[108]
		DPPH	2103.30 ± 115.60 μmol TE	
		ABTS	3037.50 ± 333.30 μmol TE	
		ORAC	4756.70 ± 41.20 μmol TE	
		TPC	305.30 ± 3.40 mg GAE/L	
	Merlot wines from Serbia and Spain * Red wine	FRAP	0.33 ± 0.01 μmol TE/g dry residue	[109]
		DPPH	0.16 ± 0.01 μmol TE/g dry residue	
		ABTS	0.35 ± 0.03 μmol TE/g dry residue	
**Coffee**	Green coffee- light roasted	DPPH	~13.00% RSA at 0.5 mg/mL sample	[110]
		ABTS	~90.00% RSA at 0.5 mg/mL sample	
	Green coffee- medium roasted	DPPH	~10.00% RSA at 0.5 mg/mL sample	
		ABTS	~90.00% RSA at 0.5 mg/mL sample	
	Green coffee- French roasted	DPPH	~6.50% RSA at 0.5 mg/mL sample	
		ABTS	~90.00% RSA at 0.5 mg/mL sample	
	Filtered coffee, water extract	TPC	13.94 ± 0.2 mg GAE/g dm	[111]
		DPPH	82.40 ± 2.86 μmolTE/g dm	
		ABTS	140.31 ± 2.80 μmolTE/g dm	
	Defatted coffee	TPC	23.43 ± 0.06 mg GAE g^−1^ dm	
		DPPH	110.33 ± 1.97 μmol TE/g dm	
		ABTS	218.38 ± 0.55 μmol TE/g dm	
**Tea**	Black Tea (Dianhong Congou)	FRAP	2670.13 ± 34.02 μmol Fe^2+^/g dw	[112]
		TEAC	994.56 ± 12.64 μmol Trolox/g dw	
		TPC	101.29 ± 1.58 mg GAE/g dw	
	Green Tea (Dianqing Tea)	FRAP	4647.47 ± 57.87 μmol Fe^2+^/g dw	
		TEAC	2532.41 ± 50.18 μmol Trolox/g dw	
		TPC	252.65 ± 4.74 mg GAE/g dw	
	Green Tea leaves	TPC	0.37 ± 0.02 mg GAE/mL at 90 °C temp	[113]
		DPPH	42.4 ± 2.6% RSA at 90 °C temp	
	Green Teabags	TPC	0.64 ± 0.02 mg GAE/mL at 90 °C temp	
		DPPH	70.3 ± 3.4% RSA at 90 °C temp	
	Black Tea leaves	TPC	0.19 ± 0.00 mg GAE/mL at 90 °C temp	
		DPPH	20.7 ± 1.5% RSA at 90 °C temp	
	Black Teabags	TPC	0.50 ± 0.02 mg GAE/mL at 90 °C temp	
		DPPH	36.0 ± 2.0% RSA at 90 °C temp	
** *Legumes* ** **Beans**	Chickpea—60% ethanol extract	TPC	21.9 ± 2.8 mg GAE/g	[101]
		TAC	648 ± 18 (U/g)	
		OH scavenging capacity	66.22 ± 0.09%	
		DPPH	~15% RSA	
	Chickpea aqueous extract	TPC	60.09 ± 4.17 mg GAE/100 g	[114]
		ORAC	52.73± 0.96 mg TE/g dry base	
		OH scavenging capacity	56.36 ± 1.54%	
**Soybeans**	The aerial part of the soybean	TPC	42.2 ± 2.23–50.40 ± 1.00 mg CE/g extract 1.40 ± 0.04 to 1.95 ± 0.00 mg CE/g fw for seven growth stages	[115]
		TEAC	177.00 ± 11.00–245.00 ± 21.00 μmol TE/g extract 6.26 ± 0.41–8.43 ± 1.28 μmol TE/g fw for seven growth stages	
		FRAP	623.00 ± 3.00–780.00 ± 0.700 μmol Fe^2+^/g extract 21.4 ± 2.6–28.5 ± 0.7 μmol Fe^2+^/g fw for seven growth stages	
		DPPH	EC_50_: 0.125–0.22 mg/mL	
	Fermented (by *M. purpureus*) defatted soybean flour	TPC	2.20 ± 0.03 mg GAE/g	[93]
		ABTS	59.61 ± 6.68 μmol TE/g	
		FRAP	14.26 ± 0.44 μmol TE/g	
		DPPH	0.74 ± 0.02 μmol TE/g	
	Water-soluble black soybean polysaccharide from sprouted seeds	TPC	3.71–6.83 mg GAE/g	[116]
		ABTS	IC_50_: 1.72–3.48 mg/mL	
		DPPH	IC_50_: 4.45–8.00 mg/mL	
		Reducing power	IC_50_: 3.42–5.84 ± 0.12 mg/mL	
** *Grains* ** **Corn**	Grounded purple corn extracted with acidified 80:20 methanol: water	TPC	9.06 ± 0.07 GAE/kg	[94]
		DPPH	IC_50_: 66.3 ± 0.80 μg/mL	
		ABTS	IC_50_: 250 ± 0.40 μg/mL	
		FRAP	26.10 ± 0.04 μmol TE/g	
	Corn	TPC	~1230–1410 μg GAE/g dm	[95]
		DPPH	37–45% RSA	
**Wheat**	Whole fresh flour	TPC	1556.11 ± 20.42 μg FAE/g	[117]
		DPPH	4.68 ± 0.45 μmol TE/g	
		FRAP	42.09 ± 2.82 μmol Fe^2+^/g	
	Wheat aleurone- water extract (WA-f50)	TPC	26.01 ± 0.40 mg GAE/g	[118]
		DPPH	147.85 ± 8.54 μmol TE/g WEAX	
		ABTS	355.26 ± 0.01 μmol TE/g WEAX	
		ORAC	527.47 ± 13.21 μmol TE/g WEAX	
	Wheat bran- water extract (WA-f50)	TPC	16.78 ± 0.35 mg GAE/g	
		DPPH	106.29± 12.13 μmol TE/g WEAX	
		ABTS	320.40 ± 21.06 μmol TE/g WEAX	
		ORAC	484.91 ± 34.15 μmol TE/g WEAX	
	Whole grain flour	DPPH	3.1 μmol TE/g	[119]
		TEAC	1.3 μmol TE/g	
		Peroxyl scavenging capacity	0.55 mmol TE/g	
	Wheat bran	DPPH	6.7 μmol TE/g	
		TEAC	2.6 μmol TE/g	
		ORAC	1.05 μmol TE/g	
**Milk and Dairy products**	Milk	ORAC_FL_	Whole milk (UHT): 14,481± 328 μmol TE Deproteinized Milk (UHT): 129 ± 5.9 μmol TE Whole milk (Pasteurized):14,216 ± 1051 μmol TE Deproteinized (Pasteurized): 464 ± 21.4 μmol TE Lowfat milk (UHT): 13,874 ± 312 μmol TE Lowfat Deproteinized Milk (UHT): 35 ± 2.2 μmol TE Lowfat milk (Pasteurized): 13,748 ± 397 μmol TE Deproteinized milk (Pasteurized):610± 16.9 μmol TE	[120]
	Yoghurt	TPC	Yoghurt; 0.14 ± 0.01 mg GAE/g (control) Yoghurt + 0.25% FSE; 0.43 ± 0.02 mg GAE/g Yoghurt + 0.5% FSE; 0.65 ±0.02 mg GAE/g	[121]
		FRAP	Yoghurt; 0.40 ± 0.03 μmol TE/g dw Yoghurt + 0.25% FSE; 2.57 ± 0.09 μmol TE/g dw Yoghurt + 0.5% FSE; 4.19 ± 0.05 μmol TE/g dw	
		ABTS	Yoghurt; 0.40 ± 0.04 μmol TE/g dw Yoghurt + 0.25% FSE; 3.63 ± 0.08 μmol TE/g dw Yoghurt + 0.5% FSE; 5.34 ± 0.23 μmol TE/g dw	

* fw: fresh weight; dw: dry weight; TE: Trolox equivalent; GAE: gallic acid equivalent; CE: Catechin equivalent Inh: inhibition; AAE: ascorbic acid equivalent; FAE: ferulic acid equivalent; RSA: radical scavenging activity; FDS: fortified with stevia extract; WEAX: water extractable arabinoxylan.

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
