# Peer review of "Oxidative Stress and Antioxidants—A Critical Review on In Vitro Antioxidant Assays"

_antioxidants, 2022, doi:10.3390/antiox11122388_

Round 1

Reviewer 1 Report

In this review, Kotha et al. summarize and describe various assays for the measurement of antioxidative capacities from various sources. In general, this review is very exhaustive and the figures greatly help to understand the topics. The tables are also of high use for everybody working in this field.

Overall the writing, phrasing and grammar of the manuscript are excellent and understandable. Very well done. Only the part 2 about oxidative stress needs a little bit of improvement (see major points for the text) and a few corrections have to be made in the manuscript itself (see minor points for the text).

If the points mentioned in detail below can be addressed by the authors in a minor revision, this already impressive review is ready for publication and will be a great contribution the field. This article represents high quality scientific writing. Very looking forward to future articles about the topic.

Author Response

Reviewer 1 comments

In this review, Kotha et al. summarize and describe various assays for the measurement of antioxidative capacities from various sources. In general, this review is very exhaustive and the figures greatly help to understand the topics. The tables are also of high use for everybody working in this field. Overall the writing, phrasing and grammar of the manuscript are excellent and understandable. Very well done.

Response: Thank you for the positive feedback and comments.

Only the part 2 about oxidative stress needs a little bit of improvement (see major points for the text) and a few corrections have to be made in the manuscript itself (see minor points for the text).

If the points mentioned in detail below can be addressed by the authors in a minor revision, this already impressive review is ready for publication and will be a great contribution the field. This article represents high quality scientific writing. Very looking forward to future articles about the topic. Major points for the text:

In Figure 1: Please remove the word ETR from the arrow, because several electron sources can deliver the electron for generating the superoxide radical. Mitochondria are only one of many sources. The reaction with ferrous iron and the superoxide generates hydroxyl radicals and Hydroperoxylradicals are rather rare and only appear in membranes.

Response: As suggested we have removed the ETR from the arrow.

In lines 78-79: Please add also one or two sentences about H2O2 with citations, since this is the most important ROS subspecies in terms of cellular signaling.

Response:  Edited as suggested

In lines 81-82: The sentence here is in several terms not correct or can be misinterpreted. Please rephrase to “ROS are oxygen-containing molecules, including radicals (like the superoxide anion) and non-radicals (like H2O2) that greatly vary in their chemical abilities, such as diffusion in living cells and chemical reactivity with biomolecules.

Response:  Edited as suggested

In lines 83-87: This part about singlet oxygen is also not fully correct and could be misinterpreted. Please rephrase to “Singlet oxygen is the highest excited spin state of molecular oxygen.

Response:  Edited as suggested

In contrast to molecular oxygen in ground state, the two valence electrons are paired in an anti-binding orbital. Singlet oxygen is therefore only generated, when molecular oxygen is energized via radiation. Importantly, and in contrast to other ROS subspecies, no electron transfer does occur during this process. Singlet oxygen is very reactive towards organic compounds and plays a deleterious role in biological systems, for instance, by involving in the oxidation of LDL cholesterol, which can lead to cardiovascular diseases.

In lines 90-91: I am very sorry, but this sentence is completely wrong. Singlet oxygen is not generated in mitochondria. You need radiation, either from a radioactive source or via sunlight (in the skin or in plants it is very common. Inside the body it is very rare and the research describing this is not very convincing). The main ROS subspecies that is generated in mitochondria is the superoxide anion. Leaking electrons are directly reacting with molecular oxygen, which is abundant in mitochondria. Therefore this passage is in high need of correction. For very recent and good overviews about ROS and their chemical behavior, please have a look at these reviews from the experts of the field: Murphy, M.P., Bayir, H., Belousov, V. et al. Guidelines for measuring reactive oxygen species and oxidative damage in cells and in vivo. Nat Metab 4, 651–662 (2022). https://doi.org/10.1038/s42255-022-00591-z Sies, H., Belousov, V.V., Chandel, N.S. et al. Defining roles of specific reactive oxygen species (ROS) in cell biology and physiology. Nat Rev Mol Cell Biol 23, 499–515 (2022). https://doi.org/10.1038/s41580-022-00456-z

Response:  Edited as suggested

In line 100: ii) “repair” does sound strange. With “repair”, in this context, normally enzymes are meant that repair the damage induced by oxidative stress, like DNA breaks, protein linkage, lipid oxidation and so on. The word “interruption” fits better. At least it’s not repair, because the process is interrupted but nothing is repaired.

Response:  Changed as suggested

In lines 108-109: In this context the word “repair of biomolecules” does not fit. You defined correctly enzymatic antioxidants as SOD, Catalase and so on. The sole function of these enzymes is to detoxify ROS. There is absolutely no repair function. They prevent the damage via removal of ROS, but they do not repair the damage already done by ROS molecules. Please rephrase.

Response:  Changed as suggested

In line 117: Please replace the “or” with an “and”. It is a commonly made misinterpretation of biologists to think that in electron delivery reactions, which involve hydrogen only the hydrogen (H+) without the electron is transferred. Since you always need to deliver an electron to transfer a radical into a nonradical the H atom is always transferred with its electron, meaning the whole hydrogen atom and not just H+.

Response:  Changed as suggested

In table 1: Is it somehow possible, to separate Advantages and Limitations in separate columns. Since this is a very nice table, this would greatly contribute to the general overview.

Response:  Changed as suggested

In line 341: Vitamin C is neither a tocopherol nor atocotrienol. Chemically it is a carbon acid. I think you wanted to place the bracelet behind Vitamin E, which would be correct.

Response:  Changed as suggested

Minor points for the text: In line 77: Please rephrase to “their function, location and amount.” Also

Response:  Changed as suggested

in line 77: You mention here for the first time the superoxide radical. Therefore the (O2-), which appears for the first time in line 88, has to be put here.

Response:  Changed as suggested

In line 89: Please write “reactions”.

Response:  Changed as suggested

In line 97: Please rephrase to “v) by the mechanisms in which they are involved [25].”

Response:  Changed as suggested

In line 99: Please write “prevention of”.

Response:  Changed as suggested

In line 110: For some exogenous antioxidants it may be not sure, but for example for Vitamin E (Tocopherol derivates) and vitamin C (Ascorbic acid) the antioxidative effects are confirmed. Therefore “may” does not fit completely.

Response:  Changed as suggested, deleted the word may

In line 120: Please rephrase to “either in vitro or in vivo”. Please also remove the “-“ between “in vitro”.

Response:  Changed as suggested

In line 203: Please rephrase to “of certain cancer types”.

Response:  Changed as suggested

In line 204: Please write “cardiovascular diseases”.

Response:  Changed as suggested

In line 251: Please remove “is a leafy green vegetable”. The habitus description is not necessary and was also not performed for the other fruits or vegetables.

Response:  Changed as suggested

In line 291: Please put a space before “According”.

Response:  Changed as suggested

In line 338: Please write “oxidative stress-related” instead of “oxidative-stress related”.

Response:  Changed as suggested

In line 351: The citation number “141” is not in bracelets. Perhaps it is a citation error.

Response:  Changed as suggested

Reviewer 2 Report

The authors gave a comprehensive review of the in vitro antioxidant assays, especially in the food aspect. Their summarization would provide researchers with some guidance on selecting proper antioxidant assays. Overall, it’s a qualified academic review and the work is well-organized and scientifically sound. Here is some suggestion for improvement of the manuscript:

1.     Figure 1, Figure 2, Figure 3, especially Figure 3 are not very clear in the printed form. Please improve the image resolution.

2.     In the abstract, the authors stated that “However, the health efficacy and the possible action of exogenous dietary antioxidants are still questionable. This may be attributed to several factors, including a lack of basic understanding of the interaction of exogenous antioxidants in the body, lack of agreement of the different antioxidant assays, and the lack of specificity of the assays, which leads to an inability to relate specific dietary antioxidants to health outcomes. “, which are the confusions in our field and the motivation of the authors to write the review. However, after the well-organized summation (Table 1 and Table 2 are good), they didn’t answer the above questions well. At the end of the manuscript, I suggest they should not only summarize the challenges but also give provide some guidance suggestions.

Author Response

Reviewer 2 comments

Top of Form

Comments and Suggestions for Authors

The authors gave a comprehensive review of the in vitro antioxidant assays, especially in the food aspect. Their summarization would provide researchers with some guidance on selecting proper antioxidant assays. Overall, it’s a qualified academic review and the work is well-organized and scientifically sound. Here is some suggestion for improvement of the manuscript:

  1. Figure 1, Figure 2, Figure 3, especially Figure 3 are not very clear in the printed form. Please improve the image resolution.

Response:  Changed as suggested

  1. In the abstract, the authors stated that “However, the health efficacy and the possible action of exogenous dietary antioxidants are still questionable. This may be attributed to several factors, including a lack of basic understanding of the interaction of exogenous antioxidants in the body, lack of agreement of the different antioxidant assays, and the lack of specificity of the assays, which leads to an inability to relate specific dietary antioxidants to health outcomes. “, which are the confusions in our field and the motivation of the authors to write the review. However, after the well-organized summation (Table 1 and Table 2 are good), they didn’t answer the above questions well. At the end of the manuscript, I suggest they should not only summarize the challenges but also give provide some guidance suggestions.

Response:  As suggested, we have added section 8 at the end entitled perspective/recommendations

Reviewer 3 Report

Dear Authors:

The manuscript "Oxidative Stress and Antioxidants - A Critical Review on in-vitro Antioxidant Assays" by Kotha et al has summarized the strengths and weaknesses of the antioxidant assays and examine the challenges in correlating the antioxidant activity of foods to human health. I have just a few suggestions.

1. Some background information or references are missing. In introduction, please add more background information about ROS, which plays an important role in many diseases. especially mitochondria damage and cancer development. It can emphasize the importance of your article. (Please cite: 1. An Epigenetic Role of Mitochondria in Cancer. Cells 2022, 11, 2518. https://doi.org/10.3390/cells11162518

  1. Advances in the Prevention and Treatment of Obesity-Driven Effects in Breast Cancers. Front Oncol. 2022 doi: 10.3389/fonc.2022.820968.
  2. mitochondrial mutations and mitoepigenetics: Focus on regulation of oxidative stress-induced responses in breast cancers. Semin Cancer Biol. 2022 Aug;83:556-569. doi: 10.1016/j.semcancer.2020.09.012.)

Best,

Author Response

Reviewer 3 comments

Comments and Suggestions for Authors

Dear Authors:

The manuscript "Oxidative Stress and Antioxidants - A Critical Review on in-vitro Antioxidant Assays" by Kotha et al has summarized the strengths and weaknesses of the antioxidant assays and examine the challenges in correlating the antioxidant activity of foods to human health. I have just a few suggestions.

  1. Some background information or references are missing. In introduction, please add more background information about ROS, which plays an important role in many diseases. especially mitochondria damage and cancer development. It can emphasize the importance of your article. (Please cite: 1. An Epigenetic Role of Mitochondria in Cancer. Cells 2022, 11, 2518. https://doi.org/10.3390/cells11162518
  1. Advances in the Prevention and Treatment of Obesity-Driven Effects in Breast Cancers. Front Oncol. 2022 doi: 10.3389/fonc.2022.820968.
  2. mitochondrial mutations and mitoepigenetics: Focus on regulation of oxidative stress-induced responses in breast cancers. Semin Cancer Biol. 2022 Aug;83:556-569. doi: 10.1016/j.semcancer.2020.09.012.)

Response:  As suggested, we have added all three references in section 2, oxidative stress section, as references 5b-5d.
